

# Characteristics of the gut microbiome in patients with prediabetes and type 2 diabetes

Zewen Zhang[*], Tian Tian[*], Zhen Chen, Lirong Liu, Tao Luo and Jianghong Dai

School of Public Health, Xinjiang Medical University, Urumqi, China
[*] These authors contributed equally to this work.

## ABSTRACT

**Background.** Gut microbiome has recently been identified as a new potential risk factor in addition to well-known diabetes risk factors. The aim of this study was to analyze the differences in the composition of gut microbiome in prediabetes(PreDM), type 2 diabetes mellitus (T2DM) and non-diabetic controls.

**Methods.** A total of 180 participants were recruited for this study: 60 with T2DM, 60 with PreDM and 60 non-diabetics (control group). Fecal samples were collected from the participants and genomic DNA was extracted. The composition and diversity of gut microbiome were investigated in fecal DNA samples using Illumina sequencing of the V3~V4 regions of 16sRNA.

**Results.** There were significant differences in the number of bacteria among patients with PreDM and T2DM and the control group. Compared with the control group, Proteobacteria bacteria were significantly higher in the PreDM group ($P = 0.006$). On the genus level, Compared with the control group, the relative abundance of Prevotella and Alloprevotella was significantly higher in the T2DM group ($P = 0.016$, $P = 0.018$), and the relative abundance of Paraprevotella in T2DM and PreDM groups was lower than that in the control group ($P = 0.011$, $P = 0.045$). Compared with the PreDM group and the control group, the relative abundance of Bacteroides in the T2DM group was significantly lower ($P = 0.019$, $P = 0.002$).

**Conclusions.** The present study found significant differences in the gut microbiome between PreDM, T2DM and non-diabetic individuals, specifically at the genus level, suggesting that early intervention in PreDM patients could have implications for gut flora transitioning to T2DM. In addition, these results may be valuable for developing strategies to control T2DM by modifying the gut microbiome.

## INTRODUCTION

Type 2 diabetes mellitus (T2DM) is a metabolic syndrome characterized by insulin dysfunction and abnormal glucose and lipid metabolism that has become one of the world's most common public health problems (*Gaike et al., 2020*). According to the statistics released by the International Diabetes Federation (*Cho et al., 2018*), in 2015 the

Corresponding author
Jianghong Dai, epidjh@163.com

global number of people aged 20–79 years with diabetes was 415 million, a number that will rise to 642 million by 2040.

The standardized prevalence of diagnosed and undiagnosed diabetes in the Chinese adult population was estimated to be 10.9% in 2013 (*Wang et al., 2017*). Prediabetes (PreDM) is defined as a condition in which blood glucose levels are higher than normal, but below the threshold for the diagnosis of diabetes (*Allin et al., 2018*). Individuals with PreDM often present overweight, with insulin resistance, and low levels of inflammation, and they suffer from an increased risk of T2DM and ischemic cardiovascular disease. It is estimated (*Wang et al., 2017*) that the prevalence of PreDM in China was 35.7% in 2013, and without intervention in the prediabetic population 70% will eventually progress to diabetes mellitus, with an annual conversion rate of 5% to 10% (*Tabak et al., 2012*). It is important to intervene proactively in the prediabetic population to interrupt or slow down the progression to T2DM. The gut microbiome has been shown to be an important factor in the development of T2DM, along with genetic, environmental, dietary, and behavioral lifestyle factors (*Gaike et al., 2020*; *Ma et al., 2019*).

The gut is the largest immune organ in the human body, and the intestinal flora residing in the gut plays a role in maintaining intestinal homeostasis, metabolism and immunity. It is also known as the "second genome" (*Lynch & Pedersen, 2016*). Healthy adult gut microbiota are dominated by *Bacteroidetes* and *Firmicutes* (>90%) but also include smaller proportions of *Actinobacteria, Proteobacteria, Fusobacteria* and *Verrucomicrobia* (*Hollister, Gao & Versalovic, 2014*; *Naseer et al., 2014*).

The gut microbiome plays an important role in regulating energy metabolism and inflammation, and is closely related to a variety of chronic diseases, such as obesity, T2DM, inflammatory bowel disease, and rheumatoid arthritis (*Festi et al., 2014*; *Komaroff, 2017*; *Lynch & Pedersen, 2016*; *Zhang et al., 2015*). Some studies have indicated that gut microorganisms directly increase intestinal uptake of monosaccharides and promote hepatic production of triglycerides associated with insulin resistance (*Larsen et al., 2010*). It has been suggested (*Sedighi et al., 2017*) that the gut microbiome can increase energy absorption from food, cause chronic low-grade inflammation, regulate fatty acid metabolism, secrete derived peptides and increase the production of metabolic endotoxins (lipopolysaccharides), leading to a chronic low rate of inflammation and insulin resistance. Previous studies have demonstrated that the quantity of *Firmicutes*, *Bifidobacteria* and *Clostridia* was significantly lower in patients with T2DM compared with that in healthy individuals (*Karlsson et al., 2013*; *Sato et al., 2014*), whereas the number of *Bacteroidetes* and beta *Proteobacteria* was significantly higher (*Qiu et al., 2019*). It was shown that the *Bacteroidetes/Firmicutes* ratio in T2DM was positively and significantly correlated with plasma glucose concentration, but appeared to be independent of body weight, confirming that it was associated with reduced glucose tolerance.

Although the causal relationship between dysbiosis of the gut microbiome and T2DM is not clear, changes in the intestinal microbiome of patients with T2DM have been confirmed (*Lambeth et al., 2015*; *Sedighi et al., 2017*). There are many reports on the intestinal microbiome and T2DM, but the results of the studies differ. T2DM treatment drugs may be one of the influencing factors. Studies have shown that diabetes treatment

drugs (such as metformin) may confuse the relationship between intestinal flora dysbiosis and T2DM (*Forslund et al., 2015*). In addition, it is not known whether there is any change in the gut microbiome of prediabetic patients and how it differs from that of T2DM and non-diabetic individuals. Few previous studies (*Gaike et al., 2020*; *Zhong et al., 2019*) have analyzed three groups of people: those with newly diagnosed T2DM, PreDM and non-diabetes. Therefore, this study recruited patients with newly diagnosed diabetes and PreDM to elucidate the characteristics of the intestinal microbiota in patients with preDM and T2DM.

## MATERIALS & METHODS

### Study population

The study was approved by Ethics Committee of the First Affiliated Hospital of Xinjiang Medical University (20191113-05). Sixty patients with newly diagnosed T2DM, 60 with preDM from the first Affiliated Hospital of Xinjiang Medical University and 60 healthy participants from the Health Management Hospital of Xinjiang Medical University were recruited for this study and signed informed consent. All participants were between 20- and 65-years-old. Healthy participants were defined as having fasting plasma glucose <5.6 mmol/L. Participants with T2DM were required to meet the following inclusion criteria: (i) fasting blood glucose test (FBG) $\geq$7 mmol/L and/or 2-h fasting oral glucose tolerance test (OGTT) $\geq$11.1 mmol/L; and (ii) all cases of T2DM were newly diagnosed. PreDM was defined as FBG of 6.1–7.0 mmol/L or HbA$_{1c}$ levels of 6.0%–6.5%. To eliminate the effects of other factors on the gut microbiota, we excluded individuals according to certain criteria: (i) age less than 20 or greater than 60 years; (ii) antibiotic usage within 2 months; (iii) habitual probiotic use; and (iv) acute and chronic gastrointestinal diseases. Information regarding demographics, diet, alcohol and tobacco use was obtained by means of a survey questionnaire. Dietary habits were assessed using a validated Food Frequency Questionnaire (FFQ). All stool samples were collected using sterile cups instantly after defecation, and sent to the laboratory within 1 hour using Styrofoam containers containing ice packs, and immediately stored in freezer at −80 °C.

### DNA extraction, PCR amplification, and 16S rRNA gene sequencing

Total DNA of microorganisms was extracted from all 180 samples using the FastDNA Spin Kit for Soil from mp biomedicals. The procedure was performed according to the instructions. DNA was quantified using a NanoDrop ND-1000 spectrophotometer (Rockland Company, USA) and stored at −80 °C for later use. The extracted genomic DNA was used to construct an amplicon library by amplifying the V3∼V4 region of the 16S rRNA gene. PCR was performed using the following conditions: initial denaturation at 95 °C for 3 min; 25 cycles with 1 cycle consisting of 95 °C for 30 s, 55 °C for 30 s, and 72 °C for 30 s; and a final extension step at 72 °C for 7 min. After the reaction, all reaction products were detected by 1.5% agar gel electrophoresis (ethidium bromide staining) to detect the amplified fragment size. A QIAquick GelExtraction Kit (QIAGEN, Germany) was used to recover and purify the target band adhesive. An Illumina Miseq high-throughput

**Table 1** Characteristics of the participants.

| Characteristics | T2DM (N = 60) | PreDM (N = 60) | Control (N = 60) | P |
|---|---|---|---|---|
| Age (years) | 49.4 ± 13.2 | 47.0 ± 14.1 | 48.5 ± 13.3 | 0.609 |
| Men/Women | 31/29 | 30/30 | 29/31 | 0.860 |
| fasting blood-glucose | 9.93 ± 3.55 | 6.43 ± 0.30 | 4.79 ± 0.47 | 0.001 |
| Smoke | 17 (28.3) | 17 (28.8) | 15 (25.4) | 0.894 |
| Drink | 8 (13.3) | 11 (18.3) | 13 (21.7) | 0.486 |

sequencing platform (Illumina) was used to sequence the PCR products in the 16S RNA V3~V4 region.

## Statistical analysis

Statistical analysis was performed using SPSS 21.0 software and R 3.43. Quality control and filtering of the raw sequencing data in order to obtain high quality sequencing data and improve the accuracy of subsequent analyses (Files S1). To analyze the differences among the groups, normally distributed variables were assessed with one-way analysis of variance (ANOVA) followed by the LSD test or Dunnett's test; categorical variables were assessed with the $\chi^2$ test. Differences in dietary frequencies among the three groups were tested using the $\chi^2$ test, and if the differences were significant, Z-test was used for pair comparison and the Bonferroni method was used to adjust for P values. The relative abundances were compared across the three groups using Kruskal-Wallis rank sum tests followed, if significant, by pair-wise comparison, and false discovery rate (FDR) using the Benjamini-Hochberg method was applied to correct the significant p-values. Alpha diversity was assessed using the Observed Species index, Chao1 index, ACE index, Shannon index, Simpson index and Coverage index. Beta diversity was assessed by principal component analysis (PCA). Two dissimilarity metrics were used: unweighted UniFrac and weighted UniFrac. Beta diversity analysis implemented in the R phyloseq pack-age. A $P$-value <0.05 was considered to be statistically significant.

## RESULT

### Group characteristics

A total of 180 participants were included in this study, with an average age of 48.7 ± 13.4 years. The distribution of age and sex was similar in the T2DM group, the preDM group and the control group. There were no statistically significant differences in smoking and drinking between the three groups ($P > 0.05$, Table 1). Analysis of the participants'FFQs showed significant differences in the intake frequency of cereal grains, mutton, eggs and products, potatoes and sweet potatoes, milk, and yogurt among participants in the three groups ($P < 0.05$). ($P < 0.05$, Table 2). here was no significant difference in the frequency of intake of rice, flour, meat, fowl, fruits and vegetables among the three study groups.

### Composition and diversity of the gut microbiome

In our study, the Observed species, Chao1 and ACE indexes were used to evaluate the richness of the microbiota, and the Shannon index, Simpson index and Coverage index

**Table 2 Dietary frequency questionnaires of study participants.**

| Categories | Dietary frequency n (%) | | | | | | P |
|---|---|---|---|---|---|---|---|
| | Control (N = 60) | | T2DM (N = 60) | | PreDM (N = 60) | | |
| | ≥1~3 times/week | Little or not | ≥1~3 times/week | Little or not | ≥1~3 times/week | Little or not | |
| Rice | 56 (93.3) | 4 (6.7) | 48 (80.0) | 12 (20.0) | 54 (90.0) | 6 (10.0) | 0.068 |
| Flour | 57 (95.0) | 3 (5.0) | 56 (93.3) | 4 (6.7) | 57 (95.0) | 3 (5.0) | 0.902 |
| Cereal (corn,sorghum, millet) | 35 (58.3)[b] | 25 (41.7) | 21 (35.0)[a] | 39 (65.0) | 22 (36.7)[a,b] | 38 (63.3) | **0.016** |
| Pork | 39 (65.0) | 21 (35.0) | 34 (56.7) | 26 (43.3) | 32 (53.3) | 28 (46.7) | 0.410 |
| Mutton | 48 (80.0)[b] | 12 (20.0) | 35 (58.3)[a] | 25 (41.7) | 38 (63.3)[a,b] | 22 (36.7) | **0.030** |
| Beef | 45 (75.0) | 15 (25.0) | 43 (71.7) | 17 (28.3) | 34 (56.7) | 26 (43.3) | 0.073 |
| Fowl | 31 (51.7) | 29 (48.3) | 23 (38.3) | 37 (61.7) | 22 (36.7) | 38 (63.3) | 0.190 |
| Seafood | 21 (35.6) | 38 (64.4) | 11 (18.3) | 49 (81.7) | 14 (23.7) | 45 (76.3) | 0.089 |
| Eggs and their product | 48 (80.0)[b] | 12 (20.0) | 34 (56.7)[a] | 26 (43.3) | 41 (68.3)[a,b] | 19 (31.7) | **0.023** |
| Offal | 5 (8.3) | 55 (91.7) | 4 (6.7) | 56 (93.3) | 2 (3.3) | 58 (96.7) | 0.483 |
| Vegetables | 59 (98.3) | 1 (1.7) | 58 (96.7) | 2 (3.3) | 59 (98.3) | 1 (1.7) | 0.786 |
| Fruits | 56 (93.3) | 4 (6.7) | 52 (86.7) | 8 (13.3) | 53 (88.3) | 7 (11.7) | 0.465 |
| Potatoes and sweet potatoes | 49 (81.7)[b] | 11 (18.3) | 33 (42.5)[a] | 27 (45.0) | 45 (75.0)[a,b] | 15 (25.0) | **0.004** |
| Beans and their product | 38 (63.3) | 22 (36.7) | 32 (53.3) | 28 (46.7) | 35 (58.3) | 25 (41.7) | 0.539 |
| Milk | 41 (69.5)[b] | 19 (31.7) | 26 (43.3)[a] | 34 (56.7) | 26 (43.3)[a] | 34 (56.7) | **0.007** |
| Yogurt (solid, liquid) | 36 (60.0)[b] | 24 (40.0) | 16 (26.7)[a] | 44 (73.3) | 14 (23.3)[a] | 46 (76.7) | **0.001** |
| Butter tea | 1 (1.7) | 59 (98.3) | 1 (1.7) | 59 (98.3) | 2 (3.3) | 58 (96.7) | 0.786 |
| Milky tea | 14 (23.3) | 46 (76.7) | 10 (16.7) | 50 (83.3) | 8 (13.3) | 52 (86.7) | 0.345 |

**Notes.**
[a,b] Comparison between subgroups at the 0.05 level after adjustment of p-values (Bonferroni method).

were used to evaluate the microbiota diversity. The results of the alpha diversity analysis showed that there were no significant differences among the three groups, including ACE, Chao1, Coverage, Observed, Shannon and Simpson indexes (Fig. 1). The bacterial community composition, assessed by principal coordinate analysis (PCoA) based on unweighted UniFrac and weighted UniFrac distances, indicated that individuals in the T2DM group and the other two groups clustered separately, presenting 31.75% and 24.2% of the total variance on the x-axis and y-axis, respectively (Figs. 2A and 2B).

Among the 180 samples tested, there were 366 different bacterial species, 217 different genera, 85 families, 27 classes and 19 phyla. The five most abundant phyla identified were *Bacteroidetes*, *Firmicutes*, *Proteobacteria*, *Actinomycetes*, and *Fusobacteria* (Fig. 3). The relative abundance of Bacteroidetes and Firmicutes was 43.3% and 45.1%, respectively, in the T2DM group, 44.9% and 41.3% in the preDM group, and 44.5% and 44.7% in the control group. Differences in relative abundance among the three groups are presented in Table 3. Compared with the control group, the abundance of phylum *Proteobacteria* was significantly higher in the preDM group (P = 0.006). *Proteobacteria* were also more abundant in the T2DM group compared with the control, though the difference was not significant (P > 0.05). Except for *Proteobacteria*, no statistically significant differences were found among the three groups at the phylum level. At the class level, only class *Negativicutes*
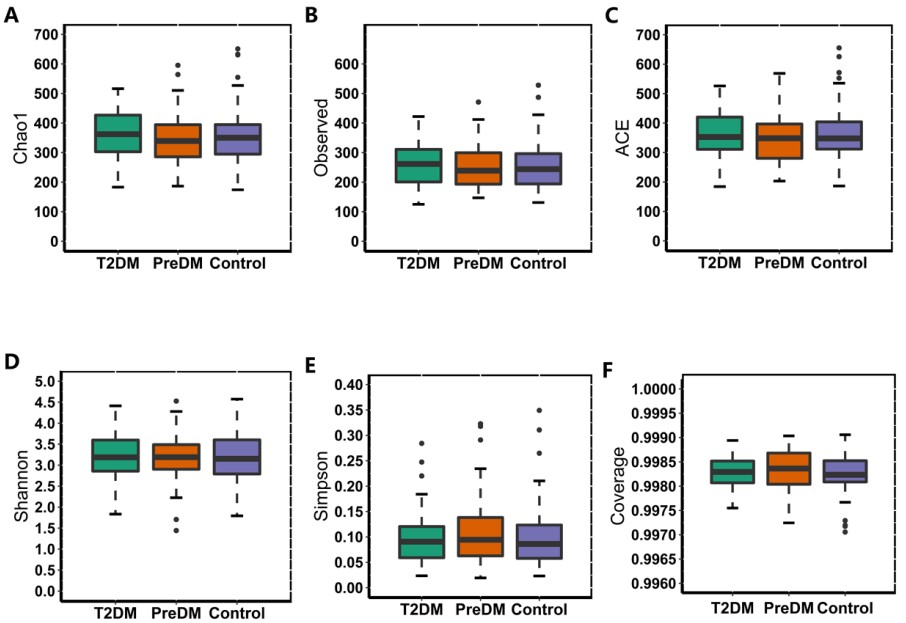

**Figure 1** Alpha diversity index of the T2DM group, PreDM group and non-diabetes group.

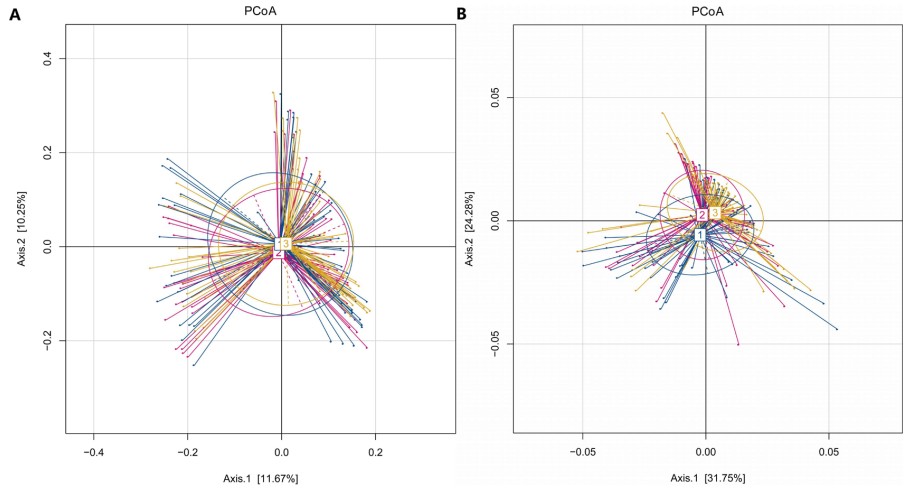

**Figure 2** PCoA of T2DM group, PreDM group and non-diabetes group. (A) Unweighted-UniFrac PCoA of T2DM group, PreDM group and non-diabetes group. (B) Weighted-UniFrac PCoA of T2DM group, PreDM group and non-diabetes group.

in the T2DM group was more abundant when compared with the other two groups, among the 27 classes (respectively $P = 0.017$, $P = 004$). Ten genera out of 217 were identified to have differences among the three groups. Compared with the control group, the relative abundance of *Prevotella* and *Alloprevotella* was significantly higher in the T2DM group ($P = 0.016$, $P = 0.018$), while genus *Paraprevotella* from phylum Bacteroidetes was less abundant in the T2DM group and preDM group than in the control group ($P = 0.011$,

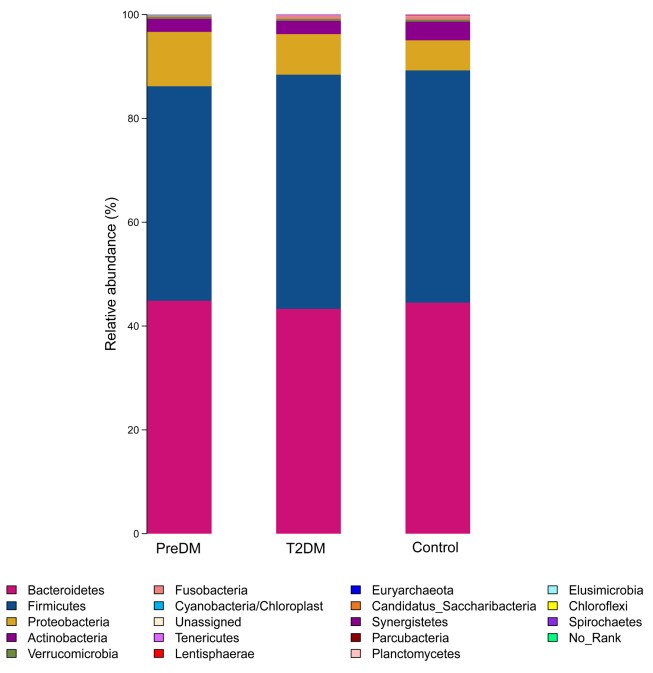

**Figure 3**  Relative richness (phylum level) in T2DM group, PreDM group and non-diabetes group.

$P = 0.045$). *Bacteroides* was found to be significantly lower in the T2DM compared with the preDM and control groups ($P = 0.019$, $P = 0.002$). *Megasphaera* was more abundant in the T2DM and preDM groups compared with control ($P = 004$, $P = 0.038$).

## DISCUSSION

A total of 180 participants (60 healthy, 60 preDM and 60 T2DM) were recruited in this study. We evaluated the diversity and compositional changes in the gut microbiota of healthy, preDM and T2DM participants. The five most abundant phyla identified were: *Bacteroides, Firmicutes, Proteobacteria, Actinomycetes,* and *Fusobacteria*, which was consistent with previous research (*Hollister, Gao & Versalovic, 2014*). Type 2 diabetes may be associated with changes in the balance of gut microbiota, but not with simple changes in the role or diversity of single microbes. *Wu et al. (2010)* compared the bacterial diversity of patients with T2DM and non-diabetic individuals, and found that there was no significant difference in bacterial diversity between the two groups, but they noticed a remarkable difference in the numbers of a few bacterial phyla, genera and species. Qin et al. conducted a study on 345 Chinese people and found no difference in microbial diversity between non-diabetic individuals and patients with T2DM; however, differences were found in composition and function, including butyrate-producing bacteria, opportunistic pathogens, and species that may reduce sulfate and degrade mucin (*Qin et al., 2012*). Our study did not find any difference between T2DM and preDM and the control group in the diversity of the gut microbiome but, when compared with the control group, both T2DM

**Table 3  Relative abundance at phylum, class and genus levels in T2DM, preDM and control groups.**

| Category | Level | Relative abundance (%) | | | P value/FDR P value | | | |
|---|---|---|---|---|---|---|---|---|
| | | ND | DM | PDM | ALL | N vs DM | N vs PDM | DM vs PDM |
| **Firmicutes** | **Phylum** | 44.7 | 45.1 | 41.3 | 0.87 | 0.8781/1 | 0.2098/0.671 | 0.1239/0.995 |
| Negativicutes | Class | 3.37 | 6.50 | 4.01 | 0.0019 | 0.0040 | 0.4635/0.833 | 0.0170/0.229 |
| Finegoldia | Genus | 0.0004 | 0.002 | 0.0005 | 0.0084 | 0.0001/0.000 | 0.3592/0.676 | 0.0050/0.075 |
| Megasphaera | Genus | 0.06 | 0.58 | 0.39 | 0.0211 | 0.0040/0.039 | 0.0380/0.181 | 0.5114/0.892 |
| Lachnospira | Genus | 0.55 | 0.48 | 0.21 | 0.6564 | 0.6534/1 | 0.0120/0.072 | 0.0949/0.499 |
| Lactobacillus | Genus | 0.60 | 1.10 | 0.38 | 0.175 | 0.2637/0.585 | 0.5345/0.823 | 0.0809/0.463 |
| **Bacteroidetes** | **Phylum** | 44.5 | 43.3 | 44.9 | 0.67 | 0.6883/1 | 0.9060/0.959 | 0.5764/0.996 |
| Bacteroides | Genus | 32.0 | 22.1 | 29.7 | 0.0014 | 0.0020/0.027 | 0.4345/0.761 | 0.0190/0.175 |
| Paraprevotella | Genus | 0.49 | 0.12 | 0.17 | 0.0075 | 0.0110/0.077 | 0.0450/0.192 | 0.4495/0.874 |
| Prevotella | Genus | 7.1 | 15.2 | 9.5 | 0.0109 | 0.0160/0.102 | 0.3856/0.690 | 0.0979/0.499 |
| Alloprevotella | Genus | 0.12 | 1.06 | 0.26 | 0.0247 | 0.0180/0.111 | 0.5124/0.816 | 0.0949/0.499 |
| **Proteobacteria** | **Phylum** | 5.8 | 7.8 | 10.5 | 0.260 | 0.1678/1 | 0.0060/0.048 | 0.2137/0.995 |
| Helicobacter | Genus | 0.0003 | 0.00004 | 0.00004 | 0.0382 | 0.0063/0.054 | 0.0156/0.089 | - |
| Escherichia/Shigella | Genus | 1.93 | 2.65 | 4.24 | 0.4848 | 0.3436/0.697 | 0.0290/0.141 | 0.2027/0.657 |
| Haemophilus | Genus | 0.11 | 0.52 | 0.48 | 0.1135 | 0.0769/0.294 | 0.0120/0.072 | 0.9240/1 |
| **Fusobacteria** | **Phylum** | 0.64 | 0.58 | 0.25 | 0.8824 | 0.9500/1 | 0.4905/0.831 | 0.3047/0.995 |
| Fusobacterium | Genus | 0.33 | 0.54 | 0.25 | 0.5391 | 0.5514/0.879 | 0.9730/1 | 0.4106/0.861 |
| **Verrucomicrobia** | **Phylum** | 0.38 | 0.37 | 0.38 | 0.98 | 0.9970/1 | 0.9590/0.959 | 0.9550/- |
| **Actinobacteria** | **Phylum** | 3.6 | 2.6 | 2.5 | 0.25 | 0.3016/1 | 0.2557/0.682 | 0.9630/- |
| Bifidobacterium | Genus | 3.1 | 1.9 | 1.9 | 0.1260 | 0.1588/0.487 | 0.1878/0.464 | 0.9690/1 |

and preDM groups had imbalance of the gut microbiome, and changes at the level of genus and class.

The complex interactions between the gut microbiome and gut mucosa may play a key role in the pathogenesis of T2DM, similar to obesity, inflammatory bowel disease and other diseases (*Bamola et al., 2017*); this may be related to an imbalance of the microbiome that may affect metabolism and cause inflammation. *Larsen et al. (2010)* found that the gut microbiome of patients with T2DM and non-diabetic individuals showed significant differences in the distribution characteristics of *Lactobacillus*, *Bacteroides*, *Clostridium*, *Bifidobacterium* and *Proteobacteria*. In patients with T2DM, Bifidobacteria, *Bacteroidetes*, and *Firmicutes* (*Lactobacillus*) were significantly lower than in the non-diabetic group, but the proportion of Proteobacteria was higher; the reason may be that, in T2DM, glucose metabolic abnormalities and increased glucose metabolites cause an increase in the numbers of pathogenic bacteria in the intestine, thus causing inflammation and insulin resistance. In contrast, however, *Sedighi et al. (2017)* found that *Lactobacillus* was significantly more common in patients with T2DM than in healthy controls, and *Bifidobacteria* were significantly lower in T2DM. In our study, compared with the healthy control group, the proportion of *Lactobacillus* was higher in T2DM, but decreased in preDM; the proportion of bifidobacteria was lower in T2DM and preDM, but no significant difference was found.
In this study, we did not find any difference in *Firmicutes* and *Bacteroidetes* among the T2DM, preDM and control groups, as also reported by *Lambeth et al. (2015)*. Firmicutes are associated with fat digestion and their increased abundance is known to be associated with obesity. *Bacteroidetes* play a key role in the production of short-chain fatty acids (SCFAs). It is also believed that *Firmicutes* and *Bacteroidetes* can enhance the absorption of monosaccharides in the gut, thus increasing the production of hepatic triglycerides and leading to insulin resistance (*Qin et al., 2012*; *Zhang et al., 2013*). However, the results for *Firmicutes* and *Bacteroidetes* in diabetic patients differ. *Ahmad et al. (2019)* found a high proportion of *Firmicutes* and a reduced number of *Bacteroidetes* in obese patients with T2DM; *Firmicutes* were enriched in T2DM and *Bacteroidetes* were found in lower numbers, resulting in a high ratio of *Firmicutes* and *Bacteroidetes* (*Navab-Moghadam et al., 2017*; *Zhang et al., 2013*); however, some other findings are contrary to this (*Larsen et al., 2010*; *Palacios et al., 2017*). Some studies have also demonstrated a significant difference in the *Firmicutes/BacTeroiDetes* ratio between thin and obese individuals (*Heinsen et al., 2016*).

We found that *Proteobacteria* and *Escherichia/Shigella* were more common in patients with preDM compared with control, and also higher in patients with T2DM, but not significantly. The outer membrane of these bacteria contains lipopolysaccharide (LPS), which is a cellular membrane component of gram-negative bacteria and is increased in both obesity and in patients with T2DM (*Sun et al., 2010*). LPS can cause metabolic endotoxemia, which is associated with oxidative stress, macrophage secreted elements and inflammatory markers that induce insulin resistance (*Momin, Bankar & Bhoite, 2016*). Previous findings have indicated that the gut microbiome of patients with T2DM is relatively rich in Gram-negative bacteria when compared with healthy individuals, especially *Proteobacteria* and *Bacteroidetes* (*Larsen et al., 2010*). In this study, we found a significantly higher abundance of the gram-negative bacterium *Haemophilus* in patients with preDM compared with healthy individuals. We also found that the T2DM group had higher levels of *Prevotella*, similar to that found in previous research (*Ahmad et al., 2019*; *Sedighi et al., 2017*; *Zhang et al., 2013*). This species is related to elevated levels of proinflammatory cytokines, low grade inflammation, and insulin resistance (*Leite et al., 2017*). In 2016, Copenhagen University and the Danish University of Science and Technology found that serum levels of branched chain amino acids (BCAAs) were increased in diabetic patients, among 277 healthy people without diabetes and 75 patients with T2DM. *Prevotella copri* and *Bacteroides vulgatus* were identified as the main species driving the association between biosynthesis of BCAAs and insulin resistance; it was found that *Prevotella copri* can induce insulin resistance, aggravate glucose intolerance and augment circulating levels of BCAAs in mice fed *Prevotella* bacteria after 3 weeks (*Pedersen et al., 2016*). However, the role of Prevotella in human gut microbiome is controversial. It was also recognized as positively associated with the production of health-promoting compounds such as short-chain fatty acids, an improved glucose metabolism or an overall anti-inflammatory effect (*De Vadder et al., 2016*; *Kovatcheva-Datchary et al., 2015*). A recent study of Italians with different dietary habits found that Prevotella's effect on diabetes was related to dietary factors and different strains (*De Filippis et al., 2019*). In addition, our study found that family Negativicutes, belonging to phylum *Firmicutes*, and *Megasphaera* were increased in both the preDM and

T2DM groups; the genera *Bacteroides* and *Paraprevotella* were reduced in patients with T2DM. Above we discussed the similarities and differences in microbial characteristics associated with T2DM and preDM, and we compared our results with those of different previous studies, as shown in Table S1. Controversial results regarding the characteristics of gut microbiome in patients with T2DM may be related to various factors, such as ethnicity, genetics, environment, geographic and climatic conditions, possible underlying diseases, lifestyle, and dietary habits of study subjects. For example, people living in different regions may have different intestinal microbial structures due to long-term differences in their dietary patterns. *Deschasaux et al. (2018)* found Moroccans, Turks and Ghanaians had a predominance of *Prevotella* in their intestines and Surinamese in Africa and South Asia had a predominance of *Bacteroides*; The subjects in this study and previous studies were located in different regions, and their dietary patterns may be quite different. For another example, differences in study methods (case control matching factors, diabetes diagnosis time, PCR specific primers and probe design, etc.) may also lead to controversial results in gut microbiome characteristics in T2DM patients. Therefore, more researches are still needed to elucidate the correct processes of gut microbiome changes associated with T2DM disease progression as well as lifestyle changes, and to gain a more detailed understanding of the role of gut microbiome composition in disease status.

There are some potential confounding factors to note when assessing gut microbiota, although attempts were made to minimize confounding variables, as much as possible, by selecting healthy controls and patients of similar age groups and sex. However, first, we lacked indicators such as height and weight to calculate participants' body mass index (BMI): the known association between BMI, obesity and gut microbiome could have affected the results (*Ahmad et al., 2019*; *Le Chatelier et al., 2013*). Diabetics are often associated with obesity, which is related to high abundance of Firmicutes and low abundance of *Bacteroides*, the diversity of gut microbiome in obese patients was significantly lower than that in normal population (*Ahmad et al., 2019*). when obese people diet and lose weight, the proportion of *Bacteroidetes/Firmicutes* will increase (*Ridaura et al., 2013*). A German study (*Thingholm et al., 2019*) showed that alpha-diversity of gut microbiome was significantly reduced in obese subjects (compared to lean healthy subjects), while there was no significant difference between obese subjects and obese T2DM. Comparing obese individuals with and without T2D showed only modest associations between the microbiome and T2D once medication and diet were also factored out, mostly characterized by a nominal increased abundance of Escherichia/Shigella. In this study, although the differences of alpha-diversity between diabetes and normal people and of Firmicutes and Escherichia/Shigella among the three groups were not detected, in the microbiome in which differences were detected, for example, Bacteroides was significantly reduced in diabetic group, possibly due to unknown BMI confounding. Second, Diet is a known factor affecting the development of the human intestinal microbiota. High intakes of carbohydrate, fat and protein are associated with increases in *Clostridium* IV and XI and decreases in the genera *Bifidobacterium* and *Lactobacillus* (*Yamaguchi et al., 2016*). In addition, studies have shown that, compared with a low dietary fiber group, the abundance of *Bifidobacterium* and *Lactobacillus* was higher in a group consuming high dietary fiber (*So et al., 2018*). Through the dietary FFQ survey, we

compared differences in the frequency of dietary intake and found that, except for a few foods, the intake frequency of most foods was not significantly different in the three groups, especially rice, meat, vegetables and fruits, which have a greater influence on the intestinal microbiome. Therefore, to a certain extent, the participants included had similar dietary habits, and this also partly reduced the influence of diet on our results. Thirdly, metformin and other drugs have been associated with changes in the gut microbiome, and studies have found that there was an increase in *Firmicutes* and decrease in *Bacteroidetes* in patients taking metformin (*Forslund et al., 2015*; *Napolitano et al., 2014*). So we included pre-diabetes and newly diagnosed diabetes, both of which generally do not use antihyperglycemic drugs, in order to reduce the impact of such drugs use in our study, and it can't deny that the lack of a medication history of the study subjects still does not completely exclude their non-use of drugs. Finally, Prediabetes are highly various groups, including impaired fasting glucose (IFG) and impaired glucose tolerance (IGT), therefore putting all preDM in one subgroup to analyse may not detect true differences in the gut microbiome profiles.

## CONCLUSIONS

In conclusion, this study reported changes in the gut microbiome associated with both preDM and T2DM, especially at the genus level. By studying the relationship between diversity and composition of the gut microbiome and metabolic diseases (such as T2DM), earlier intervention is possible to restore the microbiome to the normal state. PreDM may have an impact on the intestinal microflora in transition to T2DM, which may be altered through changes in lifestyle factors, including dietary habits and physical activity, weight management, and the use of appropriate probiotics and other substances that have a substantial impact on the gut microbiome.

### Funding
This research was supported by the Natural Science Foundation of Xinjiang Uygur Autonomous Region (No. 2018D01C144). The funders had no role in study design, data collection and analysis, decision to publish, or preparation of the manuscript.

### Grant Disclosures
The following grant information was disclosed by the authors:
Natural Science Foundation of Xinjiang Uygur Autonomous Region: 2018D01C144.

### Competing Interests
The authors declare there are no competing interests.

### Author Contributions
- Zewen Zhang analyzed the data, prepared figures and/or tables, authored or reviewed drafts of the paper, and approved the final draft.

- Tian Tian and Jianghong Dai conceived and designed the experiments, performed the experiments, authored or reviewed drafts of the paper, and approved the final draft.
- Zhen Chen performed the experiments, prepared figures and/or tables, and approved the final draft.
- Lirong Liu and Tao Luo analyzed the data, prepared figures and/or tables, and approved the final draft.

## Human Ethics

The following information was supplied relating to ethical approvals (i.e., approving body and any reference numbers):

The study was approved by Ethics Committee of the First Affiliated Hospital of Xinjiang Medical University (20191113-05).

## Data Availability

Sequencing data is available at NCBI (PRJNA661673) and Figshare:

Zhang, Zewen; Tian, Tian; Chen, Zhen (2021): Gut microbiome of diabetes. figshare. Dataset. https://doi.org/10.6084/m9.figshare.12893423.v1.

Raw data are available in the Supplemental Files.

## Supplemental Information

Supplemental information for this article can be found online at http://dx.doi.org/10.7717/peerj.10952#supplemental-information.

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
