# Peer review of "Characteristics of the gut microbiome in patients with prediabetes and type 2 diabetes"

_PeerJ, doi:10.7717/peerj.10952_

## Round 0.1 · original submission · Major Revisions

Dear authors,

Taking into account the contradictory conclusions of the reviewers I suggest performing major revisions and to discuss all remarks made by the reviewers, especially Reviewer 3.

·

Basic reporting

There is growing body of literature regarding the role of gut microbiome in the development and progression of diabetes mellitus.
The peculiarities of bacteria presence in subjects with overt diabetes mellitus and prediabetes had been shown.

Experimental design

In the manuscript gut microbiome was examined in the fecal samples in three cohorts of subjects with overt diabetes mellitus, prediabetes and control group, 60 subjects in each group.

Validity of the findings

The aim of the study and the presented data are of interest.

Additional comments

There are some questions, which I believe have to be addressed in the preparation of the final version of the manuscript.
In the Characteristics of patients studied, there are many data, which seems to be not necessary such as levels of education or income. However, there is no information about HbA1c levels and antihyperglycemic treatment if any was prescribed. This point is important as the first class of such medications, metformin, could affect gut microbiome.
The article can be shortened especially Introduction, which would be better taken by the reader.

Reviewer 2 ·

Basic reporting

English is mostly clear. However, there are some typos left.
Structure of the article corresponds to guidelines, wide range of references is used. However, I would suggest to reevaluate the Introduction section - it appears to be almost too long, some of the information has been duplicated between paragraphs (for example, lines 65 - 71). References from previous studies on microbiome of the three cohorts are missing (line 93).

Experimental design

The authors have involved and characterized a decently sized cohort of three subgroups. However, several questions need to addressed and improved regarding the data analysis.
1. Was a correction for multiple testing performed? Could you please specify the method for this and depict the adjusted values in the manuscript? Was any cofactors included in the statistical comparisons?
2. Could you please specify the tools/softwares used for beta diversity analysis?
3. What was the chosen sequencing depth?
4. Was there any data preprocessing steps performed? What tools and databases were used for that? I see some filtering info in supplementary data but the thresholds and some steps are unclear.
5. How did you ensure the immediate freezing of the samples? Were they collected in hospital environment?
6. Could you specify what method was used to disrupt the microbial cells - enzymatic or mechanistic?

Validity of the findings

The presented results are interesting and somewhat compatible with data previously reported from other studies. However, the selected cohorts unfortunately are poorly characterized and various flaws have not been discussed. For example, prediabetes ir highly various groups of patients, therefore putting all preDM patients in one subgroup could prevent from detecting true differences in the gut microbiome profiles.
In addition, despite the fact that authors acknowledge the flaw of not having any information on medications or BMI, they do not fully discuss the possibility of repeating previously characterized misinterpretation of differences between T2DM, healthy and even preDM with effects caused by antidiabetic and other(PPI, statins) medications. Could it be possible to obtain the information on medications by using some health data registries?
*it is important to note, that not having these data and not using them as cofactors for statistical analysis can lead to significant misinterpretations in the results of microbiome data comparisons!

Although various statistical comparisons have been made, the presentation of them is unclear - more precisely, it is complicated to understand at what times comparisons are performed on all subgroups together and where are specific comparisons between just two subgroups. In addition, in results section, a lot of the text is duplicating the information presented in the tables.

There is some controversy in author comments on diet data, where they characterized the diets as similar and further describe the numerous differences. I would suggest that more detailed diet data analysis is needed, most likely by inviting some nutritionist.

In discussion, it seems that results representing some controversial data to those presented by authors are missing. For example, P.copri effects on T2DM have been shown to diet and strain specific, as described by Filippis et al 2019 (https://doi.org/10.1016/j.chom.2019.01.004).

Reviewer 3 ·

Basic reporting

No comments on basic reporting

Experimental design

No comments on experimental design

Validity of the findings

A Table comparing the authors’ results with the previously published data will be very helpful and would allow to evaluate the interesting findings presented in this paper and their potential significance for the onset and escalation of the disease.

Additional comments

The present study revealed significant alterations in the gut microbiome between prediabetic (PreDM), diabetic (T2DM) and non-diabetic individuals. These interesting data suggested that early intervention in PreDM patients may have implications for gut microorganisms in transition to T2DM.
The authors cited numerous publications reporting on diabetes-related alterations in microbiomes that included some groups of microorganisms which were also analyzed in this paper and produced sometimes conflicting results (e.g., lines 212-216). While discussing significant changes found in the T2DM and pre-DM microbiomes, a Table comparing the authors’ results with the previously published data will be very helpful and would allow to better validate the interesting findings presented in this paper and their potential significance for the onset and escalation of the disease.

---

## Round 0.2 · Minor Revisions

Congratulations, now the reviewers demand minor revisions only. Please, make the necessary improvements and resubmit again your manuscript.

·

Basic reporting

The same as the in the original review.

Experimental design

The same as the in the original review.

Validity of the findings

The same as the in the original review.

Additional comments

In the revised version of the manuscript authors have addressed my previous comments.

Reviewer 2 ·

Basic reporting

no comment

Experimental design

1. The information in the manuscript and the answer to my question presents controversial information regarding the method used for microbial DNA extraction: in manuscript is says - "Total DNA of microorganisms was extracted from all 180 samples using a QIAamp stool DNA minikit (Qiagen, Germany)" (lines 125-126), however in the rebuttal letter authors have stated "DNA was extracted using the FastDNA Spin Kit for Soil from mp biomedicals.". Could the authors please clarify!

2. I would suggest to provide the sequence preprocessing descriptions with all relevant thresholds used in a supplementary file, if there is a problem with providing this info in the main manuscript (due to word limitations) - such info should not be excluded for the sake of reproducibility!

3. The authors have added the following sentence to the abstract - "..and the intestinal bacterial community was quantitatively detected by real-time fluorescence quantitative PCR." - however, there is no detailed information on such approach in the methods section. Could the authors please clarify the needed info in the methods section as well.

Validity of the findings

no comment

Additional comments

I thank the authors for extensive changes and the clarifying answers provided.

Reviewer 3 ·

Basic reporting

No comments.

Experimental design

No comments.

Validity of the findings

For most of the microbial groups examined here, the data presented in the manuscript and the results from other publications appear somewhat controversial, which should be addressed and discussed.

Additional comments

The previous comments were not completely responded to, although the Table_S1 was generated by the authors. Therefore, here are some notes for the authors:

1. Table_s1 was not referred to in Results nor it was discussed (e.g., in the paragraph l.244-251).

2. The title for Table_s1 is not appropriate and should be changed, e.g. for “Diabetes-associated differences in gut microbiome abundance: comparison with previous studies” (or “Gut microbiome abundance: comparison with previous studies”).

3. Please, make the Table more accurate and stick to a certain way of marking differences, i.e. either use arrows or “lower/higher”. Currently, the authors’ results and those by Palacios et al for Bacteriodetes, are missing from Table_s1, as well as the results by Ahmad et al., 2019 for Prevotella were not included in Table_S1.
Overall, for most of the microbial groups (except for Prevotella), the data presented here and the results from various labs appear somewhat controversial, which should be addressed and discussed.

4. The authors have added in the Introduction two new publications that reported on the same three groups of people: those with newly diagnosed T2DM, PreDM and non-diabetes, as studied in this paper (AH et al. 2020; Zhong et al. 2019; line 103) which, however, were not discussed at all in this manuscript. It would be rather appropriate if the authors could include the data from these two publications into Table_S1 and discuss them.

Correct the way the papers are cited on line 244 [ Larsen et al.(Palacios et al. 2017) found...]

---

## Round 0.3 · accepted · Accept

You have fulfilled all the demands of the Reviewers. A great job.

Reviewer 2 ·

Basic reporting

no comment

Experimental design

no comment

Validity of the findings

no comment

Additional comments

Thank you for answering the comments and improving the manuscript accordingly.